# The Use of NDT Diagnostic Methods and Calculations in Assessing the Masonry Tower Crowned with the Steel Dome

**DOI:** 10.3390/ma15207196

**Published:** 2022-10-15

**Authors:** Krzysztof Grzyb, Łukasz Drobiec, Julia Blazy, Jakub Zając

**Affiliations:** Department of Building Structures, Faculty of Civil Engineering, Silesian University of Technology, Akademicka 5, 44-100 Gliwice, Poland

**Keywords:** non-destructive technique, visual inspection, ultrasonic testing, hardness testing, historic structures, cultural heritage, steel dome

## Abstract

Non-destructive testing (NDT) methods are a diagnostic tool for evaluating the risk of failure or the need for repair and renovation. In analyzing constructions of high historical value, destructive diagnostic methods should be avoided. This study is a comprehensive NDT investigation of the masonry tower topped with a steel dome, a remnant of the overhead telecommunications network from the end of the 19th century. Visual inspection and research made it possible to assess the degree of damage to the structure. Stress–strain state analysis showed the sufficient load-bearing capacity of the steel dome. In addition, calculations have shown that the masonry tower is subjected to significant horizontal forces causing structure cracks.

## 1. Introduction

One of the most significant challenges for sustainable development and cultural heritage protection is keeping historical buildings in the best possible condition [1,2,3]. Well-preserved cultural heritage is essential, not only from a historical point of view, but because it also decides the city’s attractiveness and influences the life and safety of its habitants [4,5]. As a result, repair and renovation works are part of the life cycle of historic buildings to avoid deterioration, ageing processes, and eventual collapse [6,7]. The efficient maintenance of cultural heritage requires experience, broad knowledge, and a sense of aesthetics [8].

Non-destructive testing (NDT) methods are often used as a diagnostic tool for structures to inform about the risk of failure and the necessity of repair and renovation [9,10,11]. Their popularity in the examination of historic structures derives from the fact that NDT methods detect defects and evaluate the conditions of the materials without a necessity to destroy the studied element [12]. Additionally, NDT can be carried out not only during the exploitation of the object, but also during the construction stages to inform about its compliance with assumed requirements and properties [13].

This article presents the case study of the diagnosis and assessment of the historic post office building in Chorzów, Poland, which is around 130 years old. The methodology of the investigation process is divided into several stages, which is shown in Figure 1. The first step, while estimating the conditions and problems of the structure, is always connected with the visual assessment, which leads to the selection of the research methods. Later, the experimental investigation is provided to collect all the necessary information about the used materials and damage degree of the structure. The authors chose to use NDT methods because the studied building is a part of cultural heritage. Namely, hardness testing to check the grade of the structural steel was used because of its popularity and ease of implementation. Ultrasonic testing and microscopic methods were applied to the evolution of corrosion rate and thickness loss.

### 1.1. Visual Testing

Visual testing is the oldest known NDT method and should be carried out as the first of all the NDT methods planned for diagnostics and testing [14,15]. It allows access to the following: the condition of the external surface of the element (surface method); the presence of any discontinuities (cracks, pores, voids, undercutting, sticking); shape defects; deformations; dimensional deviations; incorrect assembly; and operational damages (corrosion, fatigue erosion, leakage) [16,17]. To determine the element’s dimensions or observed discontinuity, measuring devices such as calipers, depth gauges, and weld gauges are usually used [14]. Additionally, inspection mirrors, borescopes, endoscopes, periscopes, videoscopes, or drones may be required while diagnosing hard-to-reach places. Moreover, to observe the microstructure of the element microspores are used and when detailed analysis is needed, scanning electron microspores (SEM) are used [18,19,20]. It is also worth mentioning that the visual testing method is usually applied with some other NDT method.

### 1.2. Hardness Testing

Hardness testing distinguishes static and dynamic methods [21]. Within the former, the most popular ones are Brinell, Rockwell, and Vickers [21,22,23,24]. In all of them, the material hardness depends on force loading the indenter and permanent deformation caused by the action of this force. On the other hand, in the dynamic method, the hardness is measured based on the impact effect of the indenter on the researched surface. This method is relatively fast and easy to perform and, as a result, only minor damage to the specimen occurs, namely minor marks which are left on the tested surface [25]. In Shore’s dynamic method, a steel weight ended with a hard indenter freely falling from a certain height, and the height of reflection from the tested material is measured [16]. It is noteworthy that some dynamic methods require a minimum mass and wall thickness of the tested object [26]. In [27], Useinov et al. studied the differences in hardness results at micro and nanometer scales. On the other hand, Huber and Heerens [28] highlighted the influence of a general residual stress state on indentation and hardness testing. Finally, the effect of steel heat treatment on hardness results was observed in [29].

### 1.3. Ultrasonic Testing

Ultrasonic testing (UT) is a volumetric method that yields the detection of discontinuities inside the tested object, its size, and its location [14,18]. Thanks to this tool, it is possible to receive information about the state of its interior and microstructure [22]. A bundle of ultrasonic waves is introduced into the element [14]. Then, due to the reflection from material discontinuities, it gives indications on the screen of the ultrasonic device. Afterward, the location and size of the detected discontinuities can be assessed based on the obtained information. It must be mentioned that the UT method applies to materials with good acoustic characteristics, allowing the ultrasonic wave to propagate [30]. These are materials with an isotropic and fine-grained structure. The most dangerous internal defects in materials and welds (cracks, tears, sticking, loss of penetration) can be detected using the UT method [16]. Moreover, it is possible to measure the material’s thickness when it is accessible from only one side [16]. This is conducted by analyzing the time the wave travels through the studied element [31,32]. Additionally, modern thickness gauges yield accurate measurements without the need to remove the paint coating [33]. The ultrasonic leak testing also belongs to the UT method and gives information about the location and size of the leak. It is predominantly used to examine structures with high requirements regarding tightness, such as ships and tanks. In this method, the ultrasonic waves are emitted inside the object and if any discontinuity exists, the waves leave through them, recorded by the receiver outside the structure [34,35]. Nowadays, advanced UT techniques, such as UT phased array and UT TOFD, are more widely used. The former is based on introducing a bundle of ultrasonic waves excited in different configurations with phase shifts into the element, followed by their reception, while preserving the previous phase shifts [14]. This method is often used for elements with complex shapes and weld joints, as well as to monitor corrosion processes and create corrosion maps [36]. On the other hand, in the UT TOFD method, a bundle of ultrasonic waves is introduced to excite diffraction waves on the existing material discontinuities. Then, it is received and different flight times are measured [14]. It main application is found in examining the presence of cracks and their size in welded joints [37]. Furthermore, it is characterized by high detectability, a limited number of false readings, accurate geometry measurements of discontinuity, and a short execution time [38]. In the literature, Lin et al. proposed to use advanced UT techniques to evaluate the steel-bridge weld joints [39]. Additionally, due to its high testing speed, high efficiency, and immediately obtained results after the test, UT is one of the most commonly used NDT methods in the evaluation of steel structures, during both manufacturing and exploitation [40,41]. However, as every method of UT has its limitations and for increased roughness of the external surface of the object, the testing possibilities decrease.

A summary of the selected NDT method’s advantages and limitations, together with application indications, is shown in Table 1 [42,43].

The next stage in assessing the historic structure’s technical condition after a non-destructive diagnosis is connected to calculations and numerical analyses. There are three situations in which the exploited building may be verified. Namely, the analysis of an undamaged, damaged, or strengthened structure can be conducted (Figure 1). It must be mentioned that the steel structure has to resist forces from self-weight, live load, dead load, wind load, snow load, and accidental load. Besides this, changes in the structure’s temperature resulting from summer heating and winter cooling generate significant material stresses [44]. As a result, they must be considered while analyzing the element. Lien et al. [45] studied the non-linear behavior of steel structures subjected to heating and cooling stages. They confirmed that the elevated temperatures greatly influence the deformations of the steel frame, including deflections and displacements. Moreover, the temperature of the steel structure varies from the surrounding air temperature and can be non-uniformly distributed within the element [46,47]. Another important issue concerns the temperature at which the object was constructed because this phenomenon directly influences the thermal stresses induced in the material during exploitation [48]. Regarding the material properties, it should be considered that steel tensile and yield strength decreases with increasing temperatures [49,50]. However, it must be highlighted that both high and low temperatures have a negative influence on steel structures. Namely, when cooling down, steel becomes more brittle and losses its ductility [51,52], which is dangerous for the structure and may lead to failure. In conclusion, the problem of extreme temperatures has to be appropriately considered and addressed while analyzing steel structures.

Finally, a proper assessment of the technical condition of the structure leads to a statement about the further possibility of the functioning of the structure (Figure 1). The hazard of failure, or even collapse, should be eliminated not only for the safety of people, but also for surrounding buildings. Additionally, when it comes to the historic structure, the cultural heritage value should be taken into account. Therefore, the comprehensive assessment of the structure should be a combination of tests, measurements, engineering calculations, numerical analysis, and finally, guidelines for repair works. All of these stages were applied in assessing the technical conditions of a steel dome built on top of a brick tower of a post office in Chorzów, Poland.

The contribution of the presented case study research relies on showing the practical implementation of the theoretical background. The paper indicates the successive steps of diagnosis and evaluation of the historical building that should be taken. The analysis reveals the importance of a holistic approach during structural analysis. The article presents how experimental and numerical analysis yields the consideration of phenomena that were neglected in the past. Namely, the temperature influence was not considered while designing the steel dome built on top of the brick tower. The research significance of the work also appears in presenting specific renovation solutions for the structure. Nowadays, more and more structures designed for a 50- or 100-year service life require an assessment of the technical conditions. The article suggests the research methodology and presents a comprehensive case study.

## 2. Case Study

### 2.1. Historical Background

The building was erected in 1891–1892 in the neo-gothic style, according to the design of J. Schubert. The building was expanded in 1911 according to the design of F. Nhagen (a wing called the parcel department was added). The post office was entered into the register of monuments on 30 October 1984. Since its construction, the building has been one of the city’s landmarks.

### 2.2. Architectural Values and Building Structure

The building is a two-story building, topped with a hipped roof with dormers. The building was erected on an irregular plan. The brick tower was built on an octagonal plan, ending with the subject steel dome with a spire. The element crowning the tower is a remnant of the overhead telecommunications system from the end of the 19th century. Apart from the architectural role, the steel elements of the dome also played the role of a complex antenna system in itself, which was unique at the time of its construction. The facade is made of red clinker bricks, decorated with glazed green and brown bricks. Figure 2a shows the entrance to the post office. The binding of masonry elements with colored masonry elements, which are decorated around the window openings, is shown in Figure 2b.

The brick part of the tower is topped with a flat wooden roof. A steel hatch in the flat roof leads to the interior of the openwork structure of the dome. The construction of the cupola consists of different rod elements. The main load-bearing elements include eight steel ribs (Figure 3). The rib cross-section consists of two arches, stiffened at the ends with flat bars. The maximum width of the cross-section is 38 cm (Figure 4a), and the thickness of the rib is, on average, 15 mm. In the place where the ribs are constrained, U-shaped stiffening sheets are inbuilt. In the lower part of the dome, the ribs are connected by a decorated steel balustrade (Figure 4b). Above each intercostal segment, nine L45 × 45 angles were made. The top of the dome consists of steel pipes and flat bars that support the bell, crowning the helmet as a whole.

## 3. The Technical Condition and Diagnostic Methods

### 3.1. Macroscopically Observed Abnormalities

During on-site inspections carried out at the facility, certain damages may occur, including the corrosion of dome structure elements, incorrect connections of the dome structure elements, local constrictions of load-bearing elements, corrosion of steel built around the flat wooden roof, cracks on the tower’s masonry structure, and overgrowing vegetation on the tower body (on the facade).

Paint chips were visible on the dome’s structural elements, partly due to the corrosive processes under the paint coating (Figure 5a). The unprotected steel parts of the helmet were exposed to unfavorable weather conditions, including rainwater or moisture condensing on the steel surface. The apparent lack of proper structure protection with paint coatings led to the intensification of corrosion processes and a reduction in the effective cross-section of the element involved in the transfer of loads.

The structure inspection showed that the dome elements’ connections were mainly designed to be bolted (or riveted). In the present state, all connection points are covered with paint, and locally in some places, it is visible that the elements were welded together at the point of connections with the use of connectors (bolts/rivets). This was supposed to “strengthen” or “preserve” the connection, but such a solution cannot be considered correct (Figure 5b). Moreover, undesirable constrictions of the dome’s structural elements were locally visible, which reduces the load-bearing capacity of the bars in these places.

The roof, which separates the part of the brick tower from the steel, openwork helmet of the dome, is a wooden-beam structure. The technical condition of the flat roof can be considered reasonable; however, the dampness in the place where the rest of the beam is located is disturbing. A steel element runs around the flat roof. It was found that this highly corroded profile may suggest the accumulation of moisture in this place. This element is probably the lower circumferential brace of the dome. Figure 5c shows the high degree of corrosion of this element.

Below the flat wooden roof, numerous cracks in the masonry structure are visible from the inside of the tower. The damages are visible to the naked eye from the internal steel staircase. The cracks are vertical and horizontal (Figure 5d) and run along the circumference of the tower body. In the corners of the openings, diagonal cracks with a smaller width are visible—these are places of stress concentration resulting from lower stiffness. The damages are located below the flat wooden roof up to the upper edge of the window openings (a few meters worth of crack strips). The crack opening is at least 2.0 mm.

The abnormalities also include vegetation on the facade of the tower body, which can lead to the biological corrosion of the masonry structure or significant wall dampness in these places.

### 3.2. Non-Invasive Structure Diagnostics and Calculations

Various non-destructive diagnostic methods were used to determine the damage to the structure and establish the causes of abnormalities. Microscopic imaging enabled the assessment of the degree of corrosion of the steel elements. The analyzed areas were photographed with 50-times zoom. Such an enlargement made it possible to locate microdamage and cracks in the steel.

The actual thickness of the elements was verified using the ultrasonic method. The tests were carried out using the SONO M460 device by Metrison, Poland. The device is an ultrasonic thickness gauge designed to perform very precise, fast, and non-destructive measurements in the range of 0.65 mm to 600 mm. The device works with many heads that measure the thickness of the walls of pipes, tanks, and structural elements, both in the range of small thicknesses and those whose temperature can reach up to 300 °C, as well as corroded objects and those covered with a protective layer.

The device allows for the smooth adjustment of the ultrasonic wave speed according to the tested materials in a range from 1000 to 9999 m/s. It is also possible to use ready-programmed wave-speed settings for 22 types of materials, including the basic ones, such as steel, non-magnetic steel, aluminum, brass, copper, iron, cast iron, gold, silver, zinc, tin, nickel, titanium, epoxy resin, plexiglass, P.V.C., and ceramics. To obtain an even more accurate measurement result, the SONO M460 (Metrison, Poland) needs calibration performed by the user on samples of the measured materials. The thickness gauge used in the tests was equipped with a MEE5 5MHz ultrasonic head, which enables measurement in the range 2.5–100 mm.

The hardness tests were used to determine the actual steel grade of the cupola structure. Sauter HMM (Balingen, Germany), a type D device, was used. A rebound sensor was driven against the surface of the tested element with spring force. Depending on object hardness, the kinetic energy of the module is absorbed, and the speed reduction is measured and converted to the hardness values (Figure 6).

The voltages were proportional to the velocities and were processed and displayed as the hardness value ‘L’ (Leeb value). The surface of the tested steel should be flat with a metallic luster, and with no oxide skin or other dirt. In practice, obtaining such a surface is not easy, especially if there are corrosion losses in the steel (Figure 7). It should be added that the test sample should also be hard and stiff; otherwise, it may cause erroneous results.

Additionally, static calculations were performed on the spatial model to verify the stress state in the dome structure. The structural steel grade S235 was assumed. The structure was subjected to self-weight, wind, and climatic temperature loads. The value of the initial temperature *T*_0_ in Poland was assumed as 8 °C. The maximum heating temperature predicted in the calculations was 36 °C, and the cooling temperature was 32 °C. The calculations were also corrected, taking into account the height of the building, located at a height of 264 m above sea level. The analysis was made following the recommendations of the Eurocode. The cross-sections of the main load-bearing elements and the structure’s geometry were prepared based on an inventory made during on-site inspections (Figure 8). The loads were applied using load-panel-surface elements that transfer loads to structural elements. Calculations make it possible to assess the stress–strain state of a structure, thus determining the scope of the strengthening of structural elements.

## 4. Results

### 4.1. Results of Non-Destructive Testing

The hardness tests of structural steel in four selected places were analyzed. The result of the test was single rebound values (Leeb hardness number). The measurement results are presented in Figure 9.

Each measurement point was tested with eight approaches. The apparatus required at least five correct rebound results to determine the tensile strength and yield point correctly. If the measurement result was unreliable (gray color in Figure 9), it was not included in the estimation of mechanical parameters. The maximum standard deviation for the six rebounds was obtained at the second measuring point and amounted to 8.64. The yield point was obtained in two test points (first and second). The grade of the built-up steel was S235, although the results that were obtained suggest that the steel’s yield strength was higher and lower. Analyzing steels with alloys different from those currently used required expert assessment, particularly when applying the currently applicable standards, including assigning steels to current strength categories. Microscopic imaging made it possible to estimate the size of and damage to the structure of the steel structure. Thanks to this, it was possible to exclude cracks in the steel itself, and the cracks were mainly related to the paint coating. Moreover, remnants of another paint were discovered as shown in Figure 10.

Based on ultrasonic tests, the size of corrosion damage was estimated. In some cases, the cross-section loss reached up to 11% (Table 2).

### 4.2. Results of Calculations

The stress method was adopted, comparing the maximum stresses with the assumed structural steel yield strength. The magnitudes of stresses in the main load-bearing element ribs are shown in Figure 11 and Table 3. Stresses for the other steel elements are shown in Figure 12.

It can be assumed that the design stresses in the ultimate limit state (ULS) along the length of the bars did not exceed the steel yield stress (local exceedances exist at the point of connection of the elements with the steel ribs). Stress concentration occurs in the connection between particular elements. The structural arrangement of the dome insures the main ribs against buckling. The structure is not susceptible to a global or local loss of stability. The structure’s temperature changes, resulting from heating the elements in summer or cooling them down in winter, generating significant forces in the dome-support area. Reaction value forces in the supporting area are shown in Figure 13.

## 5. Discussion and Conclusions

The study demonstrates the usefulness of non-destructive methods in assessing a structure of unique historical value. The preliminary stage for the evaluation of the scope of the research was an on-site inspection. Then, structural steel hardness tests were carried out to assign the steel grade. The calculations made it possible to verify the load-bearing capacity of the elements.

During the local inspections, numerous cracks in the masonry tower were found. The morphology of the cracks indicated the initiation of tensile stresses on the inside of the tower. The vertical circumferential cracks suggested that forces were transmitted where the steel dome is attached. Nevertheless, damage to the masonry led to the degradation of the monument and lowered the stiffness of the entire tower structure.

These problems can result from the thermal stresses generated in the structural elements of the steel dome during intense heating or cooling. The external factors are determined by season. High stresses were, in the original form, to be taken by the circumferential steel ring connected with the main ribs of the dome. However, the existing perimeter profile in the flat roof level is highly corroded and locally cracked. Its structural function is not fulfilled. The perimeter beam’s crack should be considered a direct cause of visible cracks in the masonry structure.

Therefore, making a new ring that supports and absorbs the horizontal forces is necessary before starting repairs to the masonry structure. The tower’s masonry structure should be preserved to limit the propagation of cracks or new damage. The conducted analysis shows how important a holistic approach to structural diagnosis is. During the research, it may turn out that the observed abnormalities result from many different factors. Identifying the causes of damage to historic structures is a challenging and not obvious task. Non-destructive diagnostic methods fit in with the approach that is a compromise between preserving the complete historical form in every aspect and the need to verify the structure.

The repair of the building will consist of recreating the steel-ring top beam below the dome, which prevents the transfer of horizontal forces to the masonry structure. An additional reinforcement in the bed joints of the masonry tower structure would ensure the crack opening’s limitation.

## Figures and Tables

**Figure 1 materials-15-07196-f001:**
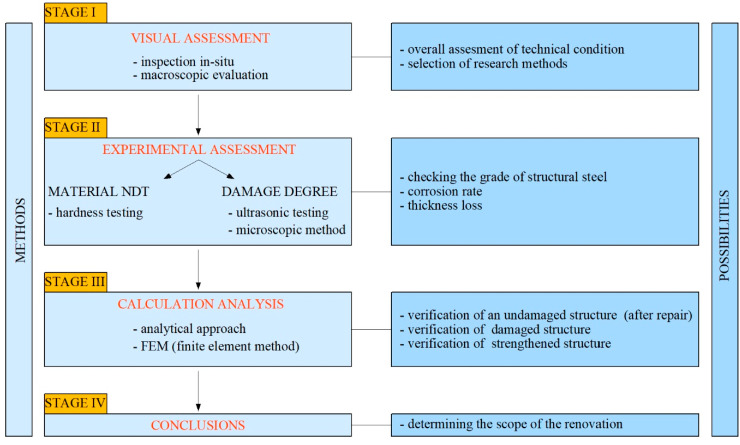
Stages of analysis of the technical condition of the structure.

**Figure 2 materials-15-07196-f002:**
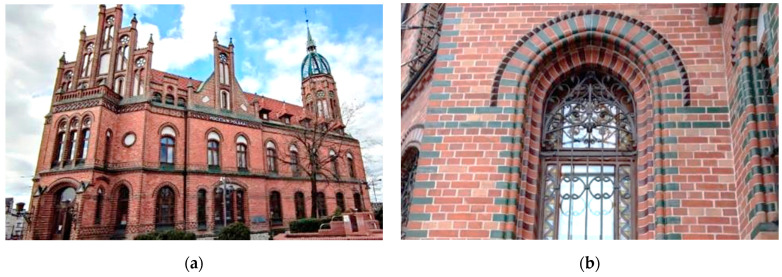
Architectonic details. (**a**) The facade of the building with a visible main entrance; (**b**) decorations in the form of colored bricks around the window opening.

**Figure 3 materials-15-07196-f003:**
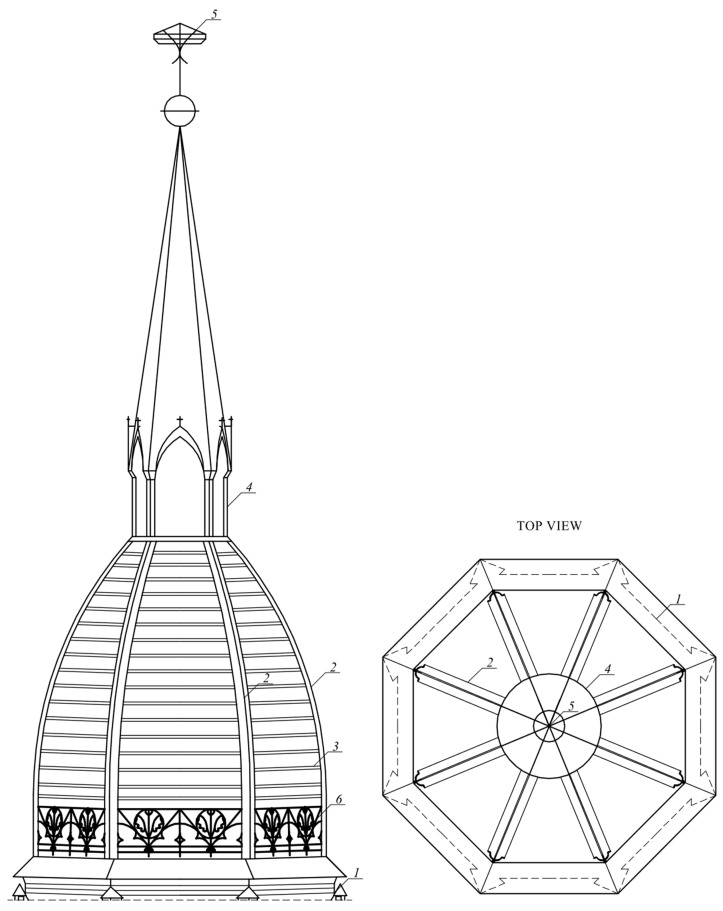
Side and top view of the dome: *1*—brick tower; *2*—ribs; *3*—angles; *4*—helmet; *5*—spire; *6*—decorative railing.

**Figure 4 materials-15-07196-f004:**
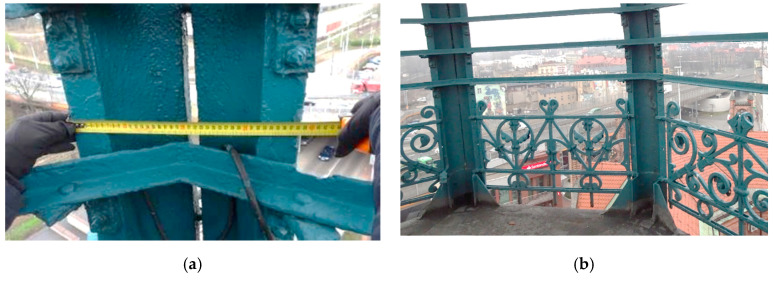
Steel dome structure. (**a**) View of the balustrade surrounding the dome; (**b**) the 38 cm width of one of the dome’s eight supporting ribs.

**Figure 5 materials-15-07196-f005:**
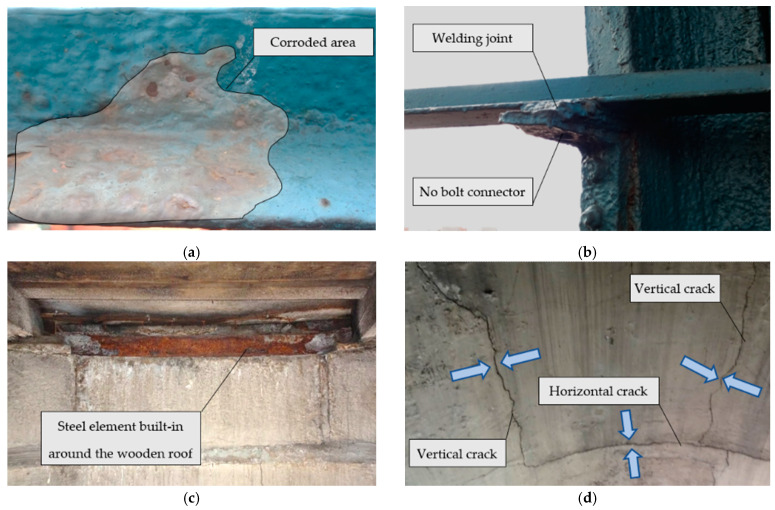
Technical problems negatively impact the structure’s technical condition. (**a**) Corrosion of the L45 × 45 angle, and visible bursting of the painted surface with corrosion products; (**b**) support for the L45 × 45 angle section at the connection with the rib; (**c**) corrosion of steel profile around the flat wooden roof; (**d**) cracks in the masonry structure below the flat wooden roof, and vertical and horizontal damages; *the arrows indicate cracks.

**Figure 6 materials-15-07196-f006:**
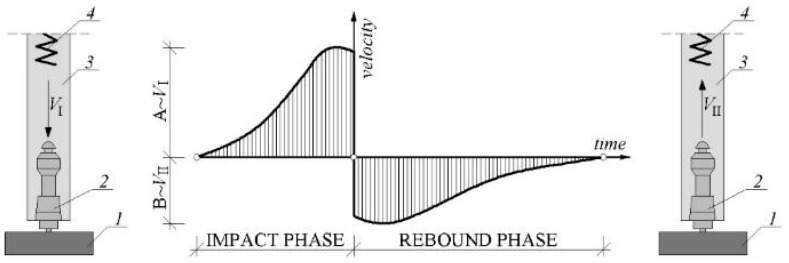
The principle of measuring the hardness of the element: *1*, tested element; *2*, movable mandrel; *3*, housing of the device; *4*, interior spring; *V*_I_, velocity in impact phase; *V*_II_, velocity in the rebound phase.

**Figure 7 materials-15-07196-f007:**
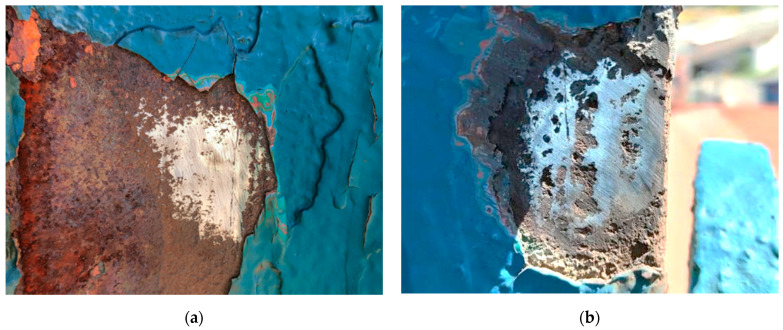
Cleaning the steel structure of paint coatings for hardness testing. (**a**) Tested area 1 of the main rib; (**b**) tested area 2 of the main rib.

**Figure 8 materials-15-07196-f008:**
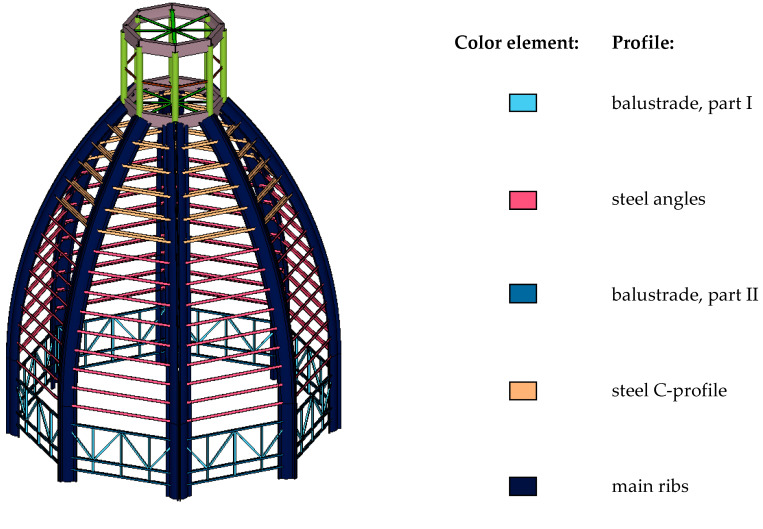
View of the spatial computational model.

**Figure 9 materials-15-07196-f009:**
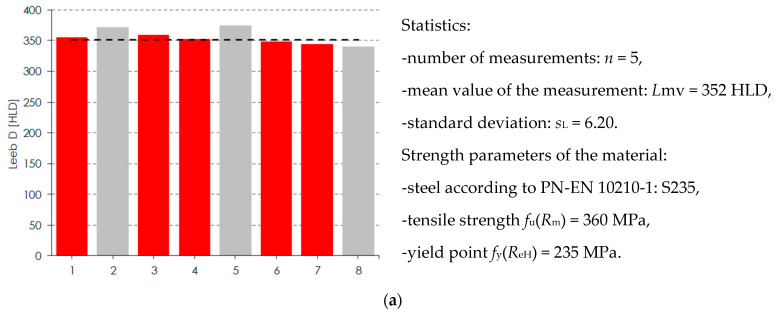
Test results for the steel structure of the dome. (**a**) Measuring point 1; (**b**) measuring point 2; (**c**) measuring point 3; (**d**) measuring point 4; gray color, unreliable rebound result omitted in the estimation of mechanical parameters.

**Figure 10 materials-15-07196-f010:**
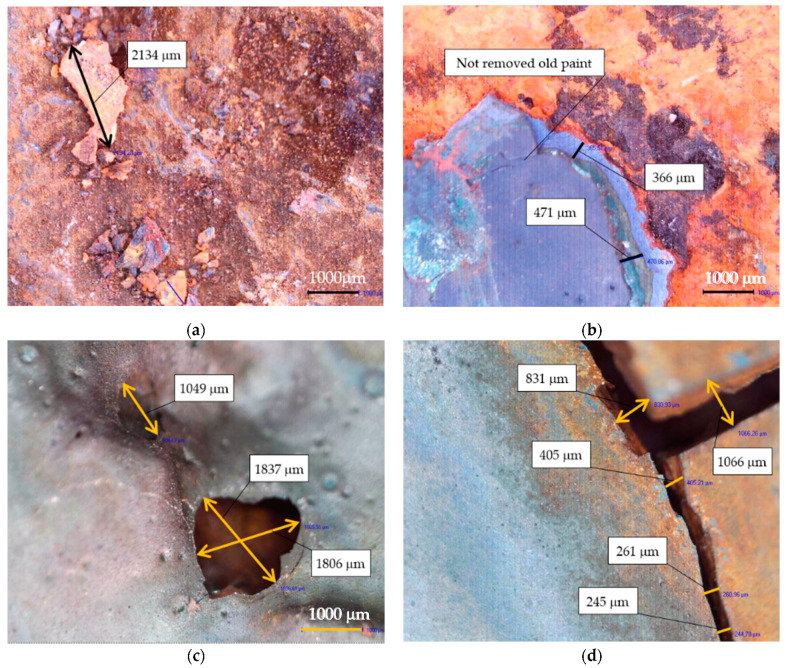
Microscopic imaging results. (**a**) Measured the size of the corrosion product 2134 µm; (**b**) thickness of the corrosion pitting 366 µm and 471 µm; (**c**) damage to paint coating, 1806 × 1837 µm size; (**d**) cracks in the paint coating, maximum size 1066 µm.

**Figure 11 materials-15-07196-f011:**
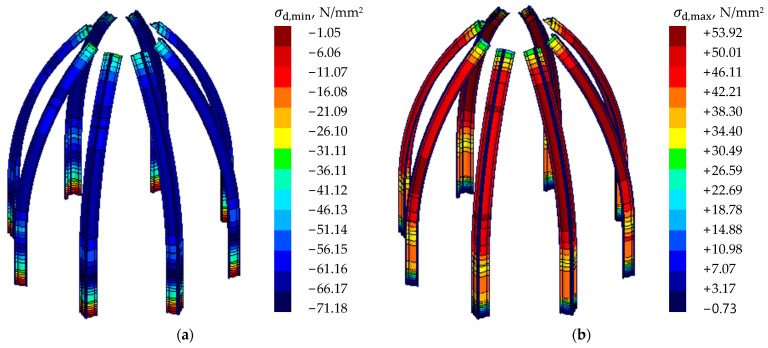
Stresses for main ribs of cupola. (**a**) minimum values; (**b**) maximum values.

**Figure 12 materials-15-07196-f012:**
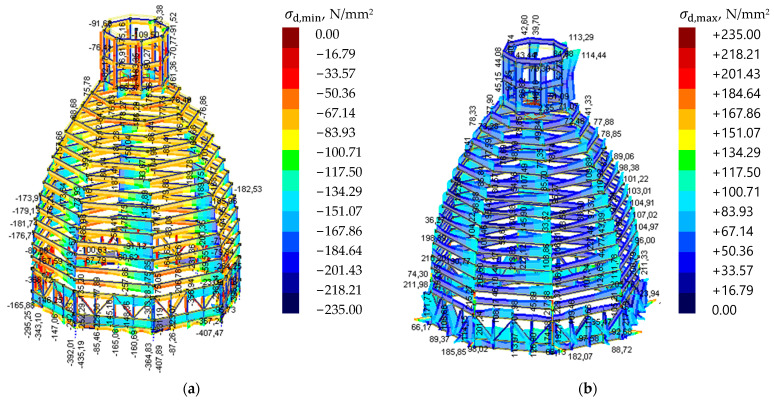
Stresses for the other steel elements. (**a**) minimum values; (**b**) maximum values.

**Figure 13 materials-15-07196-f013:**
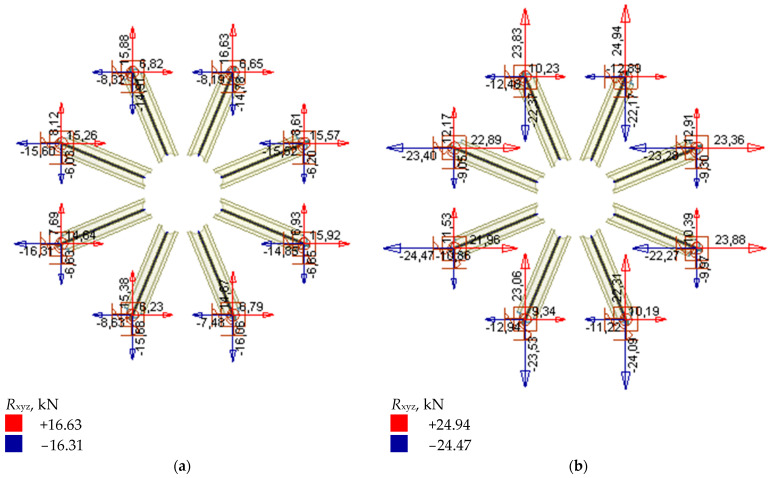
Reactions in the supporting area. (**a**) characteristic values (**b**) design values.

**Table 1 materials-15-07196-t001:** Advantages and limitations of selected non-destructive testing methods.

NDT Methods	Advantages	Limitations
Visual testing	Portable Inexpensive Commonly known Easy to implement Minimum preparation time Little or no special equipment	Only large flaws possible to detect Misinterpretations when disruptive optical effects or poor visuality
Hardness testing	Portable Commonly known Easy to implement Can be adapted to various types of materials	Requirements for surface preparation Possible deformations due to cutting and grinding Influenced by various factors
Ultrasonic testing	Portable Low costs Rapid results High sensitivity and detectability Flaw detection deep in the element Some capacity to access the location, size, nature, and shape of the discontinuity Applicable also for elements with complicated shape	Long detection time Need for coupling agent Difficult to study small, thin, and complex elements Good acoustic characteristics of tested material required For relatively smooth surfaces, limitations when roughness increases

**Table 2 materials-15-07196-t002:** Results of ultrasonic tests-measurement of the actual thickness of steel angles.

Tested Steel Angle	Element 1	Element 2	Element 3
Measured thickness range	5.98–6.00 mm	4.75–5.38 mm	5.65–5.67 mm
Average thickness	5.99 mm	5.31 mm	5.66 mm
Corrosive loss	1%	11%	6%

**Table 3 materials-15-07196-t003:** A summary of the tensile and compressive stresses of the main ribs of the dome.

Maximum Stress, *σ_d,max_*	Minimum Stress, *σ_d,min_*	Stress-Strain State, |*σ_d_*|/*f_yd_*
53.92 N/mm^2^	-	0.23
-	−71.18 N/mm^2^	0.30

## Data Availability

Not applicable.

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
