# Peer review of "The Use of NDT Diagnostic Methods and Calculations in Assessing the Masonry Tower Crowned with the Steel Dome"

_materials, 2022, doi:10.3390/ma15207196_

Round 1

Reviewer 1 Report

Comments:

This paper extensively compiled and argued the results with the comprehensive study of the state of the object should be a combination of tests, measurements, engineering calculations, numerical analysis, and finally, guidelines for repair works were analyzed.

These are my observations,

Ø  The introduction part is not explicitly written up to the scientific required.

Ø  There is need of strong literature support with justification of novelty of work.

Ø  Materials basic properties are not clearly discussed.

Ø  Add some more literature in the introduction

Ø  If possible, add a sequence of research methods.

Ø  Citations are not sequence and clear.

Ø  Conclusion needs to be re write with specific out come.

Author Response

Dear Reviewer,

We would like to thank you for taking the time to review the article and comments. As suggested, the article was supplemented and corrected according to the recommendations of the reviews:

  • The language editing of the introduction was improved.
  • The novelty of the work has been indicated.
  • The description of non-destructive material testing of the steel structure has been supplemented.
  • Literature was replenished.
  • The sequence of tests performed has been added.
  • The most important conclusions from the literature review have been collected in the new table.
  • The conclusions have been extended and completed.

We hope the introduced changes have increased the scientific value of this publication.

Reviewer 2 Report

The manuscript focuses on the use of non-destructive diagnostic methods and calculations in assessing historic structure damage, which fit with the journal's aims and scope. I think it can be considered for possible publication in the journal after a minor revision. Some recommendations are as following:

1. In the introduction, the author introduces 9 non-destructive testing methods. The advantages and disadvantages of these testing methods and their scope of application need to be further clarified.

2. How the spatial calculation model of Fig. 7 is obtained. In addition, please clarify the main role of this calculation model for structural nondestructive testing.

3. In Fig. 8, some of the test samples are 5 and some are 7. Please further explain why there are 8 data in the figure.

4. Please explain why the values of tensile strength and yield point in Fig. 8d are higher than those in Fig. 8a and 8b.

5. In Fig. 9, the blue font marked in the figure is not clear. Please modify it.

6. Whether the damage of the element is considered in the numerical calculation of Fig. 10, and why?

7. In Chapter 4, why the influence of the number of cracks, crack width and length on structural performance are not discussed? Please supplement.

Author Response

Dear Reviewer,

We would like to thank you for taking the time to review the article and comments. As suggested, the article was supplemented and corrected according to the recommendations of the reviews:

  1. The introduction has been expanded, and the advantages and disadvantages of diagnostic methods are collected in the table.
  2. Cross-sections of the main load-bearing elements and the structure's geometry were prepared based on an inventory made during on-site inspections. Calculations make it possible to assess the stress-strain state of a structure, thus determining the scope of strengthening of structural elements. NDT testing is often the starting point for numerical calculations in this approach.
  3. Each measurement point was tested with eight approaches. The apparatus requires at least five correct rebound results to determine the tensile strength and yield point correctly. If the measurement result is unreliable (gray color in Fig. 8.), it is not included in the estimation of mechanical parameters.
  4. The single rebound result is a random variable. For each measuring point, the tensile strength and yield point were assessed. The values differ because they are the results of a sample from a statistical population. Based on all the data, it was decided to classify the tested steel as S235 grade.
  5. Figure 9 has been corrected.
  6. The calculations verified the structure with cross-sections of the elements measured during the object's inventory. The calculations showed significant stocks of the bearing capacity. In this case, modeling of local corrosion losses was not necessary.
  7. The chapter Conclusion has been supplemented.

We hope the introduced changes have increased the scientific value of this publication.

Reviewer 3 Report

This paper is reporting some results on the use of non-destructive testing methods for assessing a mansory tower. This paper has no theoretical background, and the new contributions are unclear. This paper is not a research article. I think this report is not suitable for the journal. Therefore, I do not suggest this paper for publication in Materials.

Author Response

Dear Reviewer,

We would like to thank you for taking the time to review the article and comments. This paper shows the study regarding the combination of theoretical and experimental approaches. The enclosed case study of a comprehensive investigation of the masonry tower topped with a steel dome, a remnant of the overhead telecommunications network from the end of the 19th century (unique value of the world heritage of monuments of communication technology). In this work, we tried to include an extensive literature review describing the research possibilities for assessing the technical condition of monuments (the subject of the publication fits in the special issue of Materials entitled Renovation Problems in Constructions and Historic Buildings).

The reviewed version of the article has been corrected:

  • English editing,
  • An extended introduction and literature review,
  • Changes resulting from comments from other reviewers.

We hope the introduced changes have increased the scientific value of this publication.

Round 2

Reviewer 3 Report

The authors did not clearly addressed my comments. New contributions are very important for an research paper. 

Author Response

Dear Reviewer,

Thank you very much for revising the article. All changes are marked in blue color. The changed version of the article includes:

  • minor editorial corrections,
  • changing the title of the article from a general issue to a case study: The use of NDT diagnostic methods and calculations in assessing the masonry tower crowned with the steel dome
  • shortening the introduction, adjusting to the analyzed issue,
  • additional drawing describing the stages of the analysis to highlight the research significance and contribution,
  • the wording of the contribution of the article:

The contribution of the presented case study research relies on showing the practical implementation of theoretical knowledge. The paper indicates successive steps of diagnosis and evaluation of the historical building that should be taken. The analysis reveals the importance of a holistic approach during the structure analysis. It should also be mentioned that the article shows how advanced technologies support experimental and numerical analysis and allow for considering phenomena that were neglected in the past. Namely, the temperature influence was not considered while designing the steel dome built on top of a brick tower.

Nowadays, more and more structures designed for 50 or 100-year service life will require an assessment of the technical conditions. The article suggests the research methodology and presents a comprehensive case study.